# Aromatic interactions with membrane modulate human BK channel activation

**Mahdieh Yazdani[1†], Guohui Zhang[2†], Zhiguang Jia[1†], Jingyi Shi[2], Jianmin Cui[2*], Jianhan Chen[1,3*]**

[1]Department of Chemistry, University of Massachusetts, Amherst, United States; [2]Department of Biomedical Engineering, Center for the Investigation of Membrane Excitability Disorders, Cardiac Bioelectricity and Arrhythmia Center, Washington University, St Louis, United States; [3]Department of Biochemistry and Molecular Biology, University of Massachusetts, Amherst, United States

**Abstract** Large-conductance potassium (BK) channels are transmembrane (TM) proteins that can be synergistically and independently activated by membrane voltage and intracellular $Ca^{2+}$. The only covalent connection between the cytosolic $Ca^{2+}$ sensing domain and the TM pore and voltage sensing domains is a 15-residue 'C-linker'. To determine the linker's role in human BK activation, we designed a series of linker sequence scrambling mutants to suppress potential complex interplay of specific interactions with the rest of the protein. The results revealed a surprising sensitivity of BK activation to the linker sequence. Combining atomistic simulations and further mutagenesis experiments, we demonstrated that nonspecific interactions of the linker with membrane alone could directly modulate BK activation. The C-linker thus plays more direct roles in mediating allosteric coupling between BK domains than previously assumed. Our results suggest that covalent linkers could directly modulate TM protein function and should be considered an integral component of the sensing apparatus.

**\*For correspondence:**
jcui@wustl.edu (JC);
jianhanc@umass.edu (JC)

[†]These authors contributed equally to this work

**Competing interests:** The authors declare that no competing interests exist.

## Introduction

Widely distributed in nerve and muscle cells, large-conductance potassium (BK) channels are characterized by a large single-channel conductance (~100–300 pS) (*Knaus et al., 1996*; *Uebele et al., 2000*; *Salkoff et al., 2006*; *Lee and Cui, 2010*; *Yang et al., 2015*) and dual activation by both intracellular $Ca^{2+}$ and membrane voltage (*Shi and Cui, 2001*; *Horrigan and Aldrich, 2002*; *Gessner et al., 2012*), thus an interesting model system for understanding the gating and sensorpore coupling in ion channels. BK channels are involved in numerous vital physiological processes including intracellular ion homeostasis and membrane excitation, and are associated with pathogenesis of many diseases such as epilepsy, stroke, autism and hypertension (*Zhou et al., 2017*). Functional BK channels are homo-tetramers, each containing three distinct domains (*Figure 1a*). The voltage sensor domain (VSD) detects membrane potential, the pore-gate domain (PGD) controls the $K^+$ selectivity and permeation, and the cytosolic tail domain (CTD) senses various intracellular ligands including $Ca^{2+}$. The VSD and the CTD also form a $Mg^{2+}$ binding site for $Mg^{2+}$ dependent activation (*Yang et al., 2008*). The tetrameric assembly of CTD domains is also referred to as the 'gating ring'. VSD and PGD together form the trans-membrane domain (TMD) of BK channels. Previous studies on mouse BK channels (*Yang et al., 2008*) and recent atomistic structures of full-length *Aplysia californica* (aSlo1) and human BK channel (hSlo1) (*Hite et al., 2017*; *Tao et al., 2017*; *Tao and MacKinnon, 2019*) reveal that CTD of each subunit reside beneath the TMD of the neighboring subunit in a surprising domain-swapped arrangement (*Figure 1—figure supplement 1a*).

The only covalent connection between CTD and TMD of BK channels is a 15-residue peptide referred to as the 'C-linker' (R329 to K343 in the human BK channel, hSlo1) (green in *Figure 1a*). This

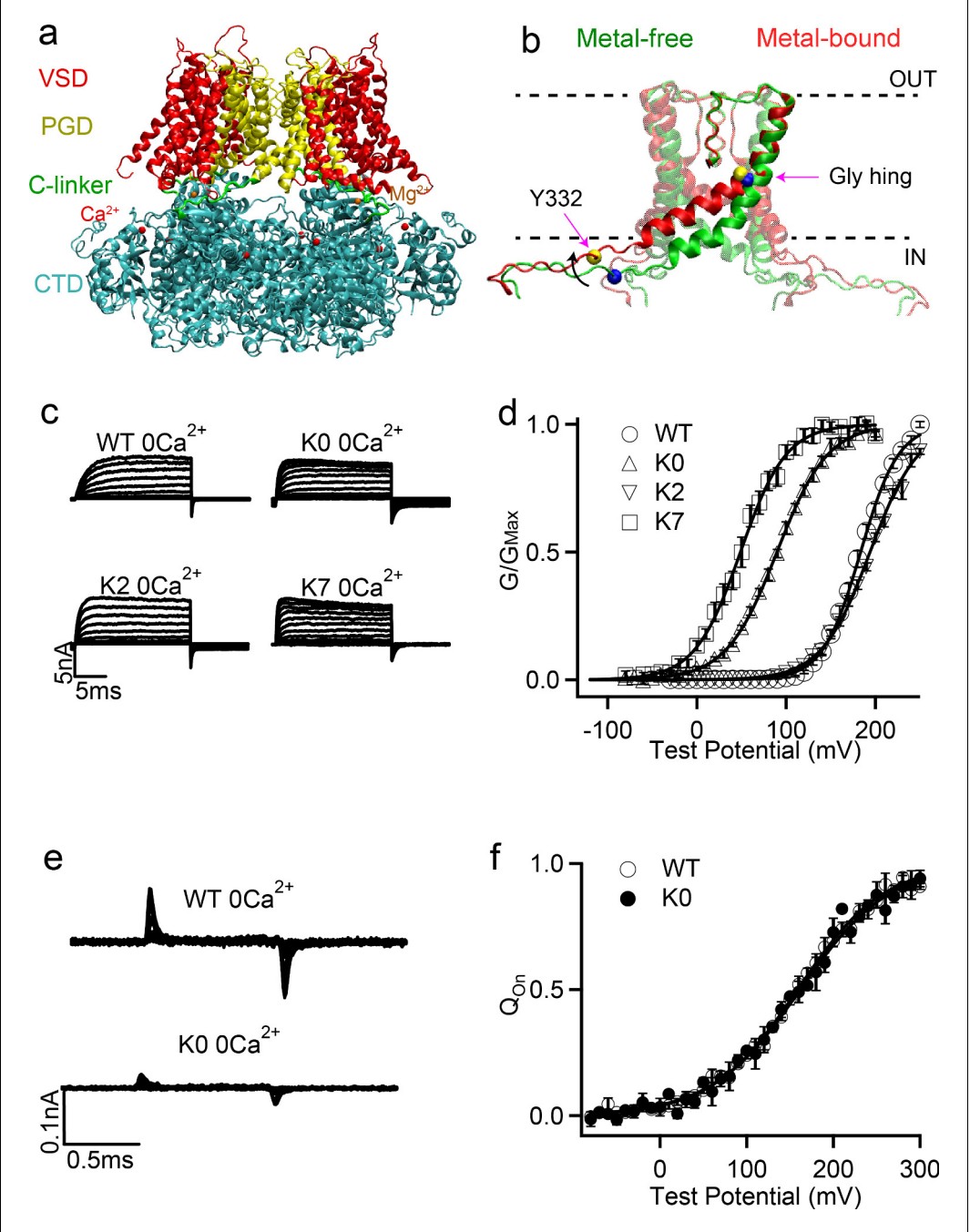

**Figure 1.** Structure of BK channels and effects of C-linker sequence scrambling on its voltage activation. (**a**) Key functional domains and structural organization of the human BK channel (hSol1). See main text for definition of domains; (**b**) Orientation of the pore lining S6 helices and C-linkers in the metal-free (green) and metal-bound (red) states; The dash lines indicate the approximate positions of membrane interfaces. The location of Tyr 332 and Gly 310 are indicated by yellow and blue spheres in the metal-bound and metal-free states, respectively. The black arrow shows S6 movement upon metal binding. (**c**) Macroscopic currents of WT, K0, K2 and K7 hSlo1 channels. The currents were elicited in 0 $[Ca^{2+}]_i$ by voltage pulses from −30 to 250 mV with 20 mV increments for WT and K2 and voltage pulses from −80 to 200 mV with 20 mV increments for K0 and K7. The voltages before and after the pulses were −50 and −80 mV, respectively. (**d**) Conductance-voltage (G-V) curves for WT, K0, K2 and K7 hSlo1 channels in 0 $[Ca^{2+}]_i$ showing significant shifts in the activation voltage $(V_{0.5})$; All solid lines were fit to the Boltzmann relation (see Materials and methods), with $V_{0.5}$ of 183.4 ± 3.2 mV for WT; 89.6 ± 3.5 mV for K0; 195.5 ± 3.5 mV for K2; and 48.7 ± 4.7 mV for K7. (**e**) Gating current traces of WT and K0, Voltage pulses were from

*Figure 1 continued on next page*

*Figure 1 continued*

−30 to 300 mV (WT) or from −80 to 300 mV (K0) with 20 mV increments. (**f**) Normalized gating charge-voltage (Q-V) relation of on-gating currents. The smooth curves are fits to the Boltzmann function with a $V_{0.5}$ and slope factor of $159.1 \pm 6.5$ mV and $49.0 \pm 5.9$ mV for WT, and $161.9 \pm 9.2$ mV and $51.6 \pm 8.5$ mV for K0.

The online version of this article includes the following source data and figure supplement(s) for figure 1:

**Source data 1.** Data from electrophysiology experiments showing G-V curves for WT, K0, K2 and K7 hSlo1 channels in 0 [$Ca^{2+}$] as depicted in *Figure 1d*.

**Source data 2.** Data from electrophysiology experiments showing Q-V relation of on-gating currents for WT and K0 as depicted in *Figure 1f*.

**Figure supplement 1.** Structural properties of BK channels.

**Figure supplement 2.** Packing between C-linker and VSD/RCK1 N-lobe of WT hSlo1.

**Figure supplement 3.** Comparison of the aSlo1-derived homology model of hBK (orange) with the new Cryo-EM structure (PDB 6V38) (cyan) in the metal-bound state.

**Figure supplement 4.** Structural and dynamic properties of hSlo1 obtained from simulations of the recently published Cryo-EM structures (PDB: 6V3G and 6V38).

**Figure supplement 5.** A representative structural model of hSlo1 with 12 residues (3x AAG units; colored red) inserted after S337 in the C-linker region (colored green).

---

linker directly connects the pore lining S6 helices in the PGD (yellow in *Figure 1—figure supplement 1b*) to the N-terminus of CTD (known as RCK1 N-lobe; blue in *Figure 1—figure supplement 1b*), and is believed to play an important role in mediating the gating ring-pore coupling (*Gessner et al., 2012*; *Yang et al., 2015*; *Zhou et al., 2017*). For example, (*Niu et al., 2004*) observed that lengthening the C-linker through inserting poly-AAG (*Supplementary file 1*) was accompanied with right shifted voltage-dependent activation measured by conductance-voltage relations (G-V curves), while shortening the C-linker led to left-shifted G-V curves, clearly demonstrating the importance of C-linker in BK gating. Intriguingly, the voltage required for half activation, $V_{0.5}$, displayed a highly linear relationship with the number of residues inserted or deleted in the absence of $Ca^{2+}$. This led to the proposal that the linker-gating ring behaves as a 'passive spring' in activation of BK channels (*Niu et al., 2004*).

Evidence has also accumulated to suggest that the C-linker may play more direct roles in mediating allosteric coupling of BK channels. Both $Ca^{2+}$-free and bound Cryo-EM structures of full-length aSlo1 (*Hite et al., 2017*; *Tao et al., 2017*) were found to contain wide-open pores and thus did not provide clue to how the channel might be gated. Atomistic simulations later revealed that, due to the movement of pore-lining S6 helices (*Figure 1b*), the BK pore cavity become narrower, more elongated, and crucially, more hydrophobic in the metal-free state (i.e., without bound $Ca^{2+}$ and $Mg^{2+}$; presumably the closed state) (*Jia et al., 2018*). As such, the pore can readily undergo hydrophobic dewetting transition to give rise to a vapor barrier that prevents ion permeation (*Jia et al., 2018*). Recognition of hydrophobic gating in BK channels provides a mechanistic basis for further understanding how the C-linker may mediate the sensor-pore coupling. Specifically, key movements of S6 helices involve bending at the glycine hinge (G310, G311) toward the membrane upon $Ca^{2+}$ binding (*Figure 1b*), leading to ~6 Å expansion of the pore entrance near I323. It has also been proposed that BK channels may be gated at the selectivity filter (*Wilkens and Aldrich, 2006*; *Kopec et al., 2019*; *Schewe et al., 2019*), the conformation of which could be modulated by the S6 helix orientation. The precise gating mechanism of BK channels thus remains to be further established. Nonetheless, considering that C-linkers are directly connected to S6 helices (*Figure 1—figure supplement 1b*), their interactions with the rest of the channel as well as membrane and water will likely have an effect on the S6 orientation and consequently BK channel activation. The notion that interactions of C-linkers can modulate BK activation has actually been demonstrated in a recent study, where the $R_{329}KK_{331}$ residues in the C-linker was proposed to form alternating interactions with E321 and E224 from the neighboring chains and membrane lipids during each gating cycle (*Tian et al., 2019*).

A key challenge of using mutagenesis to delineate the roles of a specific residue or interaction in protein function is that multiple competing effects may be perturbed simultaneously. The C-linker, in particular, is involved in extensive interactions with the gating ring and VSD (*Figure 1—figure supplement 2*), making it very difficult to derive unambiguous interpretation of its role in BK activation

(*Tian et al., 2019*). In this work, we examine a set of BK channels with scrambled C-linker sequences to determine if the C-linker is largely inert in mediating the sensor-pore coupling of BK channels. Combining mutagenesis, electrophysiology and atomistic simulation, we discover that the C-linker is a major pathway of gating ring-pore communication and can play a much more direct and specific role in mediating BK activation than previously thought. In particular, we show that dynamic interactions of the C-linker with the membrane/solvent environment, namely, membrane anchoring effects, can directly modulate voltage gating of BK channels. The dynamic and nonspecific nature of these interactions is in contrast to specific lipid-protein interactions, which involve binding of one or more lipid molecules to well-defined pockets on the protein in specific poses, or specific protein-protein interactions, which involve certain pairs of residues confined in space by the folded structure of the entire protein. This observation has been verified by additional experiments performed on BK constructs either lacking CTD or with the membrane anchoring residue mutated.

## Results

### Sequence scrambling of the C-linker dramatically modulates BK voltage activation

Studies have shown that splice variant encompassing the C-linker region would profoundly affect the gating characteristics of BK channels (*Soom et al., 2008*). Although informative, such studies usually modify not only the composition but also the length (number of amino acids) in the C-linker region, making it difficult to draw firm conclusions about the role of the C-linker on the gating characteristics. Motivated by the linker gating 'passive spring' model (*Niu et al., 2004*), we aimed to investigate such behaviors through designing a series of BK mutants where the C-linker sequence has the same set of amino acids but the ordering is scrambled (*Table 1*). This design would allow exploring the 'passive spring' model without eliminating or adding amino acid. If the linker-gating ring largely acts like a passive spring with an inert C-linker, the expectation is that these scrambling mutant BK channels would have similar gating properties. *Table 1* shows experimental results on the scramble mutant channels. Among these mutants K3, K5 and K6 did not show functional expression, while other mutant channels showed robust currents (*Figure 1c*). We measured voltage-dependent activation of these channels and the conductance-voltage (G-V) relationships were fitted using the

**Table 1.** C-linker scrambling mutations and measured $V_{0.5}$ in the full-length at both 0 [$Ca^{2+}$] and 100 µM [$Ca^{2+}$] and Core-MT BK channels at 0 [$Ca^{2+}$].

The Core-MT constructs are based on the TMD, C-linker of mSlo1, and an 11-residue tail from $K_V$ 1.4 of the mouse Shaker family (*Budelli et al., 2013*; *Zhang et al., 2017*). The location of the nearest Tyr to the S6 C-terminal is underlined. K0 (Y330G) was designed to remove the Tyr sidechain in the K0 background.

| Mutation | Sequence | $V_{0.5}$ (mV) | | Core-MT |
|---|---|---|---|---|
| | | **Full-Length** | | |
| | | 0 [$Ca^{2+}$] | 100 [$Ca^{2+}$] | 0 [$Ca^{2+}$] |
| WT | EIIEL IGNRK K<u>Y</u>GGS YSAVS GRK | 183.4 | 0.2 | 235.0 |
| K0 | EIIEL IGNR<u>Y</u> GKGSK YSRAV SKG | 89.6 | −66.7 | 192.6 |
| K0(Y330G) | EIIEL IGNR<u>G</u> GKGSK YSRAV SKG | 169.8 | 47.5 | NM |
| K1 | EIIEL RIGNK <u>Y</u>GGSY KSAVR KSG | 136.2 | −4.2 | NM |
| K2 | EIIEL IGRKN <u>Y</u>KGGS YSARV SGK | 195.5 | 59.0 | 263.5 |
| K3 | EIIEL IGN<u>Y</u>G GRSYS KAKVS RKG | NC | NC | NM |
| K4 | EIIEL IGRN<u>Y</u> GGSYS AKKVR SKG | 94.6 | −50.4 | NM |
| K5 | EIIER LIGKK RN<u>Y</u>KG GSYSA VSG | NC | NC | NM |
| K6 | EIIEL RKKIR KGN<u>Y</u>G GSYSA VSG | NC | NC | NM |
| K7 | EIIEL IGN<u>Y</u>G GSYSA VRKSK GRK | 48.7 | −63.7 | 167.5 |
| | NC: no current, channel could not be expressed; NM: Mutation Not Made | | | |

Boltzmann function to derive $V_{0.5}$, the voltage where $G/G_{Max}$ reaches 0.5 (*Figure 1d*). Left shift of G-V ($V_{0.5}$ decreases) indicates that the channel requires less voltage to activate, while right shift of G-V ($V_{0.5}$ increases) indicates that the channel is opened by higher voltage (See Materials and methods for details). Importantly, as shown in *Figure 1d* and *Table 1*, $V_{0.5}$ depends very sensitively on the C-linker sequence, which is in line with previous observations (*Soom et al., 2008*). All five mutants that lead to functional channels, K0, K1, K2, K4 and K7, have significantly altered activation voltage, with $V_{0.5}$ changes as large as ~135 mV.

We further examined whether the C-linker scrambling mutation alters voltage-dependent gating by altering activation of the VSD. We measured the gating currents of the K0 mutant in BK channels (*Figure 1e*). Gating currents are generated by the movement of VSD during voltage-dependent activation, and the gating charge movement at various voltages, the Q-V relationship, of the K0 mutation is compared to that of the WT channels (*Figure 1f*). The results show that the K0 mutation gives rises to Q-V curves similar to that of the WT channels. This is in contrast to the G-V relation of the K0 mutation, which is shifted to more negative voltages by about −95 mV (*Table 1*). These results show that the K0 mutation does not affect voltage sensor movements, but may alter pore opening to modify BK channel activation.

## C-linker is a structured loop with limited flexibility and a key dynamic pathway of BK gating ring-pore coupling

Atomistic modeling and simulation were performed to investigate possible mechanisms of unexpected sensitivity of BK gating on the C-linker sequence. We first derived atomistic structures of the wild-type (WT) hSlo1 based on the aSlo1 Cryo-EM structures (*Hite et al., 2017*; *Tao et al., 2017*) (see Materials and methods). The homology models have been cross validated using Cryo-EM structures of hSlo1 that became available after the completion of this work (*Tao and MacKinnon, 2019*). The results show that aS1o1-derived models are essentially identical to the actual hSlo1 structures, with the central PGD structures differ by only 0.87 Å (*Figure 1—figure supplement 3*). Atomistic simulations also suggest that the homology models and Cryo-EM structures lead to similar structural and dynamical properties (e.g., see *Figure 1—figure supplement 4*). These structures reveal that the C-linker is a structured loop with essentially identical conformations in both metal-free (closed) and metal-bound (open) states (*Figure 1b*). The $C_\alpha$ root-mean-square deviation (RMSD) of C-linker conformations between the two states is only ~0.8 Å. The linker forms extensive contacts with the RCK1 N-lobe (H344 to N427) of the gating ring, mainly mediated by the $Y_{332}GGSYSA_{338}$ segment in the C-linker, and S0' of VSD (*Figure 1—figure supplement 2*). In particular, the two conserved tyrosine residues (Y332 and Y336) are fully embedded in a hydrophobic RCK1 N-lobe pocket, apparently maintaining a tight packing between the C-linker and RCK1 N-lobe (*Figure 1—figure supplement 2*). Several positively charged residues (R329, K330, K331, R342 and K343) flank the above segment and are exposed to solvent, likely mediating the solvation of the C-linker. Importantly, these two short tracks of charged residues appear to be tightly anchored by the C-linker-RCK1 contacts.

The stability of the C-linker conformation as a structured loop is further confirmed by atomistic molecular dynamic (MD) simulations, which showed that the C-linker maintained stable conformations and contacts with VSD and RCK1 N-lobe throughout the 800 ns simulation time (*Figure 2a–b*, *Figure 2—figure supplement 1*, *Figure 2—figure supplement 2*, *Figure 2—figure supplement 3* and *Figure 2—figure supplement 4*, WT). The root-mean-square fluctuation (RMSF) of the C-linker region is ~1–2 Å, which is only slightly elevated compared to regions with helical or sheet secondary structures (*Figure 2b*). The end-to-end distance of the C-linker fluctuated stably ~30 Å in both metal-bound and free state simulations (*Figure 2a*), even as the pore underwent dewetting transitions in the metal-free state (*Figure 2—figure supplement 4c*).

Dynamic network analysis further reveals that the C-linker is a key pathway of dynamic coupling between the gating ring and PGD. Such analysis utilizes correlation of residue motions during MD simulations to uncover probable pathways of allosteric coupling in biomolecules (*Eargle and Luthey-Schulten, 2012*; *LeVine and Weinstein, 2014*; *McClendon et al., 2014*). The optimal and suboptimal paths of coupling are then identified as the shortest paths with the highest pairwise correlations, which should possess the highest probabilities of information transfer (*Eargle and Luthey-Schulten, 2012*). We analyzed the optimal and suboptimal pathways of coupling between I323 in the S6 helices, where substantial conformational change occurs during the gating event (*Jia et al., 2018*), and critical metal binding residues, including D895 and R514 in the RCK2 and RCK1 $Ca^{2+}$

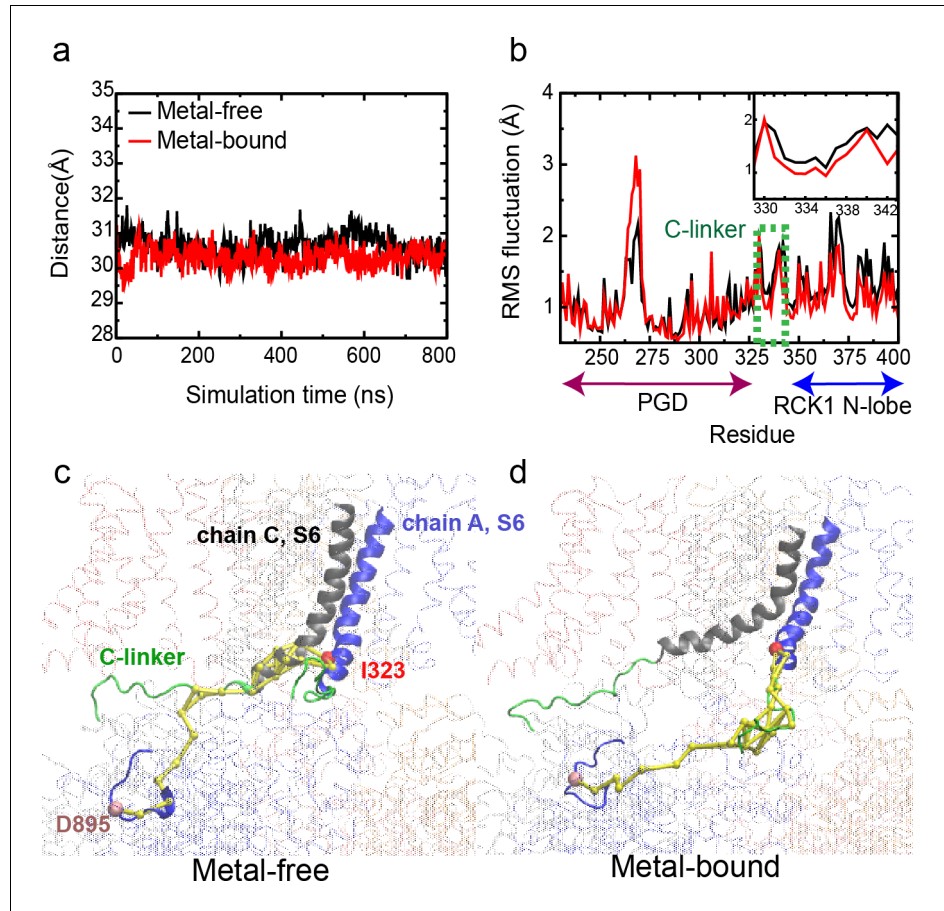

**Figure 2.** Dynamic properties of BK channels. (**a**) The C-linker N-C Cα distance during a representative 800-ns MD simulations of the WT hSlo1. (**b**) Residue Root-Mean-Square Fluctuations (RMSF) profiles of PGD, C-linker and RCK1 N-lobe derived from the same trajectory (green dashed box and shown in the insert). (**c-d**) Optimal and top 10 suboptimal pathways of dynamic coupling (yellow traces) between D895 in the RCK2 $Ca^{2+}$ binding site of chain A ($C_\alpha$ colored as pink) and I323 in PGD of chain A ($C_\alpha$ colored as red) in the metal-free (**c**) and bound (**d**) states. The C-linker is colored green, S6 of chain A and its neighboring chain (chain C) in blue and grey, respectively. The rest of the channel is shown as transparent ribbons. Note that the pathways in the metal-free state go through the neighboring chain (chain C).

The online version of this article includes the following source data and figure supplement(s) for figure 2:

**Source data 1.** Data obtained from simulation studies (see Materials and method section structural and dynamic analysis) calculating C-linker N-C Cα distance for both metal-bound and metal-free states of WT hSlo1 channels as depicted in *Figure 2a*.

**Source data 2.** Data obtained from simulation studies (see Materials and method section structural and dynamic analysis) calculating residue RMSF for both metal-bound and metal-free states of the WT hSlo1 channels as depicted in *Figure 2b*.

**Figure supplement 1.** Probabilities of residue-residue contacts between the C-linker and VSD of the neighboring chains.

**Figure supplement 2.** Probabilities of residue–residue contacts between the C-linker and RCK1 N-lobe of the same chain.

**Figure supplement 3.** Structural and dynamic properties of the C-linker in WT hSlo1 and mutants.

**Figure supplement 4.** Structural and dynamic properties of WT hSlo1 and C-linker mutants.

**Figure supplement 5.** Optimal (red) and suboptimal (green) dynamic coupling pathways between the critical residues in the $Ca^{2+}$ and $Mg^{2+}$ binding sites (Cα colored as purple sphere) and I323 (Cα colored as yellow sphere) in PGD.

**Figure supplement 6.** Correlation between the average intrinsic end-to-end distance of free C-linkers and measured $V_{0.5}$ of the WT hSlo1 and mutants.

binding site, respectively, and E374 in the $Mg^{2+}$ binding site residing in the CTD. The results reveal that the main pathways of communication from $Ca^{2+}$ and $Mg^{2+}$ binding sites to the PGD, go through the C-linker for all four chains (*Figure 2c–d* and *Figure 2—figure supplement 5*). This is not necessarily surprising since C-linker is the only covalent connection between the domains. Interestingly though, in the metal-free state the main path from the RCK2 $Ca^{2+}$ binding site to PGD goes from the neighboring monomer in every other chain (*Figure 2c–d*). This can be attributed to much tighter S6 helix packing in the metal-free state (*Jia et al., 2018*), and, combined with the domain swapped arrangement of BK tetramers, may help enforce cooperative gating response upon metal binding.

## C-linker scrambling mutations minimally perturb channel structure, dynamics and sensor-pore coupling

Atomistic simulations suggest that the changes in voltage dependent gating upon permutating linkers are unlikely to derive from a change in the overall structural features and dynamical properties of BK channel. The structure of the channel appears minimally perturbed by the mutated linkers, with the overall RMSD below ~5 Å and the TM domain (core) RMSD around 2–3 Å from the initial cryo-EM-derived structures for both WT and mutant channels (*Figure 2—figure supplement 4a–b*). All mutant channels can readily undergo hydrophobic dewetting transitions as observed for the WT channel (*Figure 2—figure supplement 4c*). The linker region also maintains similar backbone conformations in WT and all mutants (*Figure 2—figure supplement 3*), even though it becomes slightly more dynamic in the K7 mutant as reflected in the RMSF profile (*Figure 2—figure supplement 3b*). Furthermore, scrambling mutations do not appear to perturb long-range coupling properties either; the C-linker remains to provide the key pathway of dynamic coupling between CTD and PGD.

Sequence properties, particularly distributions of charged residues, can modulate intrinsic conformational properties of disordered peptides including chain extension (*Das and Pappu, 2013*), which could explain the changes in channel activation in terms of the linker-gating ring passive spring model. To investigate this hypothesis, we performed atomistic simulations of the isolated C-linker segments in the ABSINTH implicit solvent (*Vitalis and Pappu, 2009*), which has been shown to reliably predict the inherent conformational extension of disordered peptides (*Mao et al., 2013*). The results suggest that $V_{0.5}$ of linker scrambling mutants is not correlated with the inherent linker extension (*Figure 2—figure supplement 6*) and is inconsistent with the prediction based on the previous linker-gating ring passive spring model (*Niu et al., 2004*) (dashed line in *Figure 2—figure supplement 6*). Therefore, these inherent C-linker properties do not dictate the functional changes as we observed, which support the idea that the C-linker still adopts the same conformation as the WT C-linker shown in the structures (*Figure 2—figure supplement 3*). These results also indicate that some interactions other than the inherited properties of the peptides are the cause of our experimental observations. As shown later, it is the aromatics-membrane interactions that are responsible for our experimental observations.

Finally, since the C-linker is a structured loop with extensive interactions with the CTD and VSD (*Figure 1—figure supplement 2*, *Figure 2—figure supplement 1* and *Figure 2—figure supplement 2*), we considered whether some specific subsets of mutations could perturb the VSD to modulate voltage-dependent activation ($V_{0.5}$), even though sequence scrambling is designed to suppress such effects. It has been proposed that interactions of lipid headgroups with charged patches of VSD stabilize the down state of the VSD in the Kv1.2 channel (*Delemotte et al., 2011*). To test this, we analyzed C-linker residue contact probability maps from MD trajectories to identify potential group of contacts that could be correlated to $V_{0.5}$ (*Figure 2—figure supplement 1* and *Figure 2—figure supplement 2*). Contact analysis show that the C-linker does not have any contact with the putative gating charges of BK VSD (R207, R210 and R213) and thus unlikely interfere with potential lipid-VSD interactions. This is consistent with the experimental results that the K0 mutation does not alter VSD movements during voltage-dependent activation (*Figure 1f*).

## Tyr membrane anchoring effects can modulate S6 orientation to affect BK activation

As discussed above, key conformational changes involved in activation of BK channels include the re-orientation of the pore-lining S6 helices, which bends outwardly and toward the membrane at the glycine hinge (G310 and G311) (*Figure 1b*; *Jia et al., 2018*). Directly connected to the S6 helix,

C-linker residues would be moved closer to the membrane interface during activation (e.g., comparing red vs green cartoons in *Figure 1b*). As such, it can be anticipated that nonspecific interactions of the C-linker with the membrane interface can affect channel activation by stabilizing the bent conformation of S6 helices, thus modulating the activation voltage. Among various amino acids, aromatic ones such as Tyr and Trp are known to be 'membrane anchoring', with a strong preference toward localizing at the membrane-water interface (*Johansson and Lindahl, 2006*; *Monticelli et al., 2008*). While the sequence scrambling was designed to suppress the potential effects of specific interactions of the C-linker, the two Tyr residues (Y332 and Y336 in WT) were placed at different separations from the membrane interface (*Table 1*, underlined). Indeed, the measured $V_{0.5}$ shows a strong correlation with the sequence separation between E324 (the approximate location of membrane interface) and the nearest C-linker Tyr residue (*Figure 3b*) in both 0 $[Ca^{2+}]_i$ and 100 μM $[Ca^{2+}]_i$. Positioning this Tyr residue closer to the membrane interface could allow stronger membrane anchoring, preferentially stabilizing the bent conformation of S6 in the open state and shifting the equilibrium toward the active state of the channel as observed. This effect is not affected by the change of $[Ca^{2+}]_i$, suggesting that the interaction between the Tyr residue and the membrane does not alter $Ca^{2+}$ dependent activation.

To more directly examine if membrane anchoring plays a role in modulating BK activation, we further analyzed the atomistic trajectories to understand the details of Tyr interaction with the membrane interface. As illustrated in *Figure 4*, Tyr sidechains contain both aromatic rings and polar groups that allow them to embed their aromatic rings in the lipid tail region and at the same time direct the polar -OH groups toward the lipid headgroup region to form hydrogen bonding interactions with water and lipid headgroups (*Johansson and Lindahl, 2006*). In addition, the aromatic ring could also form π-cation interactions with positively charged cholines in lipid headgroups (*Roberts et al., 2018*). Importantly, results from simulation analysis confirm that positioning of the C-linker in the metal-bound state allows more extensive interactions between Try sidechains and the

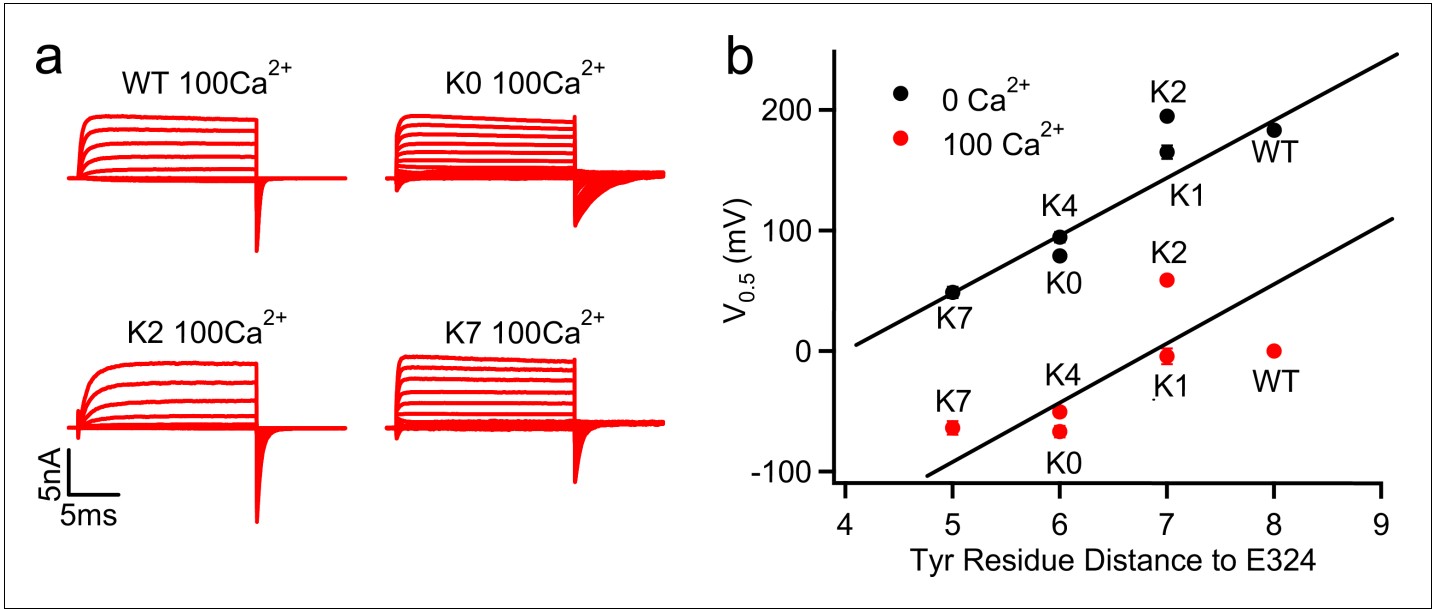

**Figure 3.** G-V shifts caused by the C-linker mutations correlated with Tyr position. (**a**) Macroscopic currents of WT, K0, K2 and K7 mSlo1 channels in 100 μM $[Ca^{2+}]_i$. The currents were elicited by voltage pulses from −200 to 100 mV with 20 mV increments. The voltages before and after the pulses were −50 and −120 mV, respectively. (**b**) G-V shifts as a function of the sequential distance (residue counts) of the nearest C-linker Tyr (Y332 in WT; Y330 in K0; Y331 in K1; Y331 in K2; Y330 in K4; Y329 in K7) to the S6 C-terminal (E324). $V_{0.5}$ values were obtained by fitting Boltzmann equation to G-V relations in 0 (black) (*Figure 1d*) and 100 μM (red) $[Ca^{2+}]_i$. Straight line for 0 $[Ca^{2+}]_i$ data is a linear fit to the data to highlight the correlation, for the 100 μM $[Ca^{2+}]_i$ data is a parallel shift of the line for 0 $[Ca^{2+}]_i$.

The online version of this article includes the following source data for figure 3:

**Source data 1.** Data from electrophysiology experiments showing the relation of the G-V curves to the position of the nearest C-linker Tyr as depicted in *Figure 3b*.

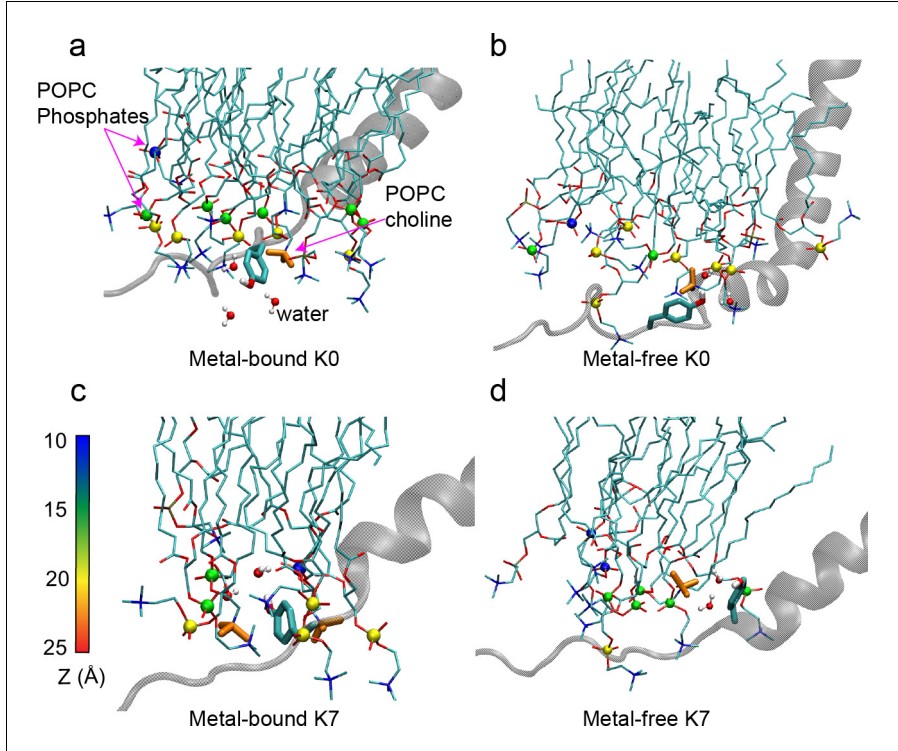

**Figure 4.** Interactions of the C-linker Tyr residue nearest to the S6 helix with the membrane interface in hSlo1 K0 (Y330, panels **a, b**) and K7 (Y329, panels **c, d**) mutants. Only the S6 helix and C-linker from one subunit are shown for clarity. POPC molecules near the Tyr residue are shown in sticks, with the phosphorous atoms shown in spheres and colored according to their distance to the membrane center (Z). POPC choline groups (orange Licorice) and water molecule near the Tyr sidechain are also shown to illustrate the π-cation and hydrogen bonding interactions. Note that the C-linker is positioned closer to the interface and forms more extensive interactions in the metal-bound (activated) state.

membrane interface (e.g., compare *Figure 4a* vs b and c vs d), which would stabilize the open state and thus lower the voltage required for activation. This is more clearly shown in the average hydrophobic, hydrogen-bonding and π-cation interactions formed by the nearest Tyr sidechains (Y332 in WT, Y330 in K0, Y331 in K2, and Y329 in K7; see *Table 1*), summarized in *Figure 5* and *Figure 5—figure supplement 1*. This is consistent with the observation that Tyr membrane anchoring lowers $V_{0.5}$ significantly for all mutants except K2. Note that Y329 of K7 appears to be dominated by extensive π-cation interactions in the metal-bound state (*Figure 5d*, comparing empty and filled bars) and as a consequence forms slightly fewer hydrogen bonding interactions on average (*Figure 5b*). Consistent with our experimental data, placing Tyr closer to the end of the S6 helix shifts $V_{0.5}$ (*Figure 1c–d* and *Figure 3*) indeed allows more extensive Tyr/membrane interface interactions, forming larger numbers of polar and nonpolar interactions on average (*Figure 5*). However, we note that directly quantifying the free energy contribution of Tyr membrane anchoring effects is technically difficult. It requires rigorous calculation of the free energy of the open/close transition of the whole channel, which is not yet feasible given the current computational capability. In particular, there is substantial heterogeneity in the lipid distribution near the channel, especially around the C-linker region (e.g., see *Figure 4* and *Figure 5—figure supplement 2*). Achieving convergence on the free energy of activation transition would require sufficient sampling of these lipid configurations, which has been shown to be extremely slow at the multi-μs level or longer (*Neale et al., 2014*).

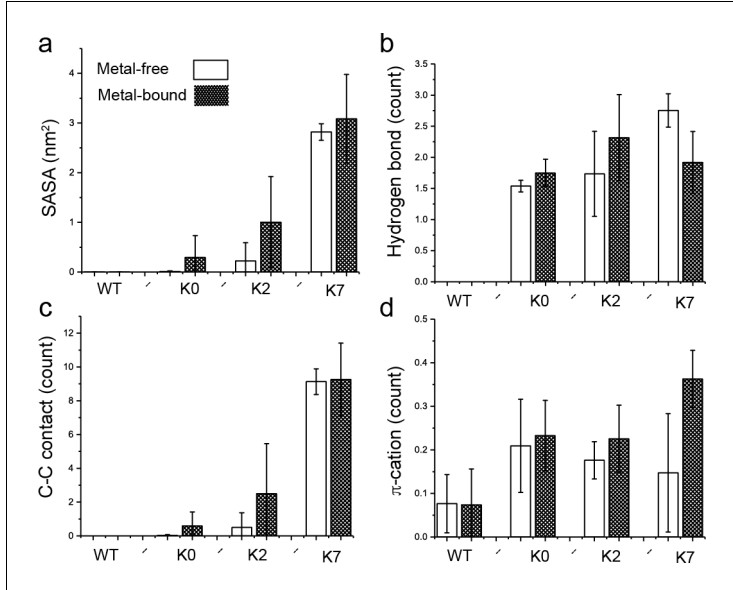

**Figure 5.** Interactions of the C-linker Tyr residue nearest to the S6 helix with the membrane interface. (a) Average Tyr sidechain solvent accessible surface area (SASA) of burial by lipid tails, representing the level of hydrophobic contacts between the Tyr sidechain and aliphatic lipid tails. (b) Average number of hydrogen bonds between the Tyr OH group and the POPC hydrophilic head groups. (c) Average number of carbon-carbon (C-C) contacts between the Tyr aromatic ring and POPC hydrocarbon tails. (d) Average number of π-cation interactions between the Tyr aromatic ring and POPC choline group. All results are the average of three independent simulations, with the standard error shown as the error bar. No hydrophobic, hydrogen bonding, or C-C contacts as defined above were observed in the WT channel.

The online version of this article includes the following source data and figure supplement(s) for figure 5:

**Source data 1.** Data extracted from simulation studies (see Materials and method section structural and dynamic analysis) of WT, K0, K2 and K7 comparing different interactions of Tyr with the membrane components as depicted in *Figure 5a–d*.

**Figure supplement 1.** Distributions of various interactions of the C-linker Tyr residue nearest to the S6 helix, with the membrane interface for WT hSlo1 and mutants derived from simulations.

**Figure supplement 2.** A representative snapshot of metal-bound K0 channel showing membrane distortion around the protein.

## Tyr membrane anchoring affects BK channel activation similarly without the gating ring

If the observed effects on the voltage gating of full-length BK channels are indeed mainly due to the nonspecific interactions between the C-linker and membrane interface, they should persist in the Core-MT truncated channel (*Budelli et al., 2013*; *Zhang et al., 2017*), in which the whole gating-ring is removed and there is no CTD coupling with either VSD or PGD via the C-linker (*Budelli et al., 2013*; *Figure 6a*). To test this prediction, Core-MT channels with three C-linker scrambling mutations, K0, K2 and K7, were expressed (*Figure 6b*) and their voltage-dependent activation measured. Mutations of the C-linker shifted the G-V relation of the Core-MT constructs in the same directions as observed for the full-length channels (*Table 1*), with K0 and K7 stabilizing channel activation (*Figure 6c,e*) while K2 making activation at higher voltages (*Figure 6d*). However, the reduction in the activation voltage by K0 mutation is only ~42 mV in Core-MT, compared to ~94 mV in the full-length channel (*Figure 6c*), and for K7 mutation is ~67 mV in Core-MT, compared to ~135 mV in the full-length channel (*Figure 6e*). This is not considered surprising as the C-linker is more flexible in the absence of the gating ring, thus weakening the effects of membrane anchoring. We measured the gating currents of the Core-MT and the K0 mutant in Core-MT (*Figure 6f*). The Q-V curves for these two channels are similar (*Figure 6g*) supporting that the K0 mutation does not affect voltage sensor movements, but may alter pore opening to modify BK channel activation.

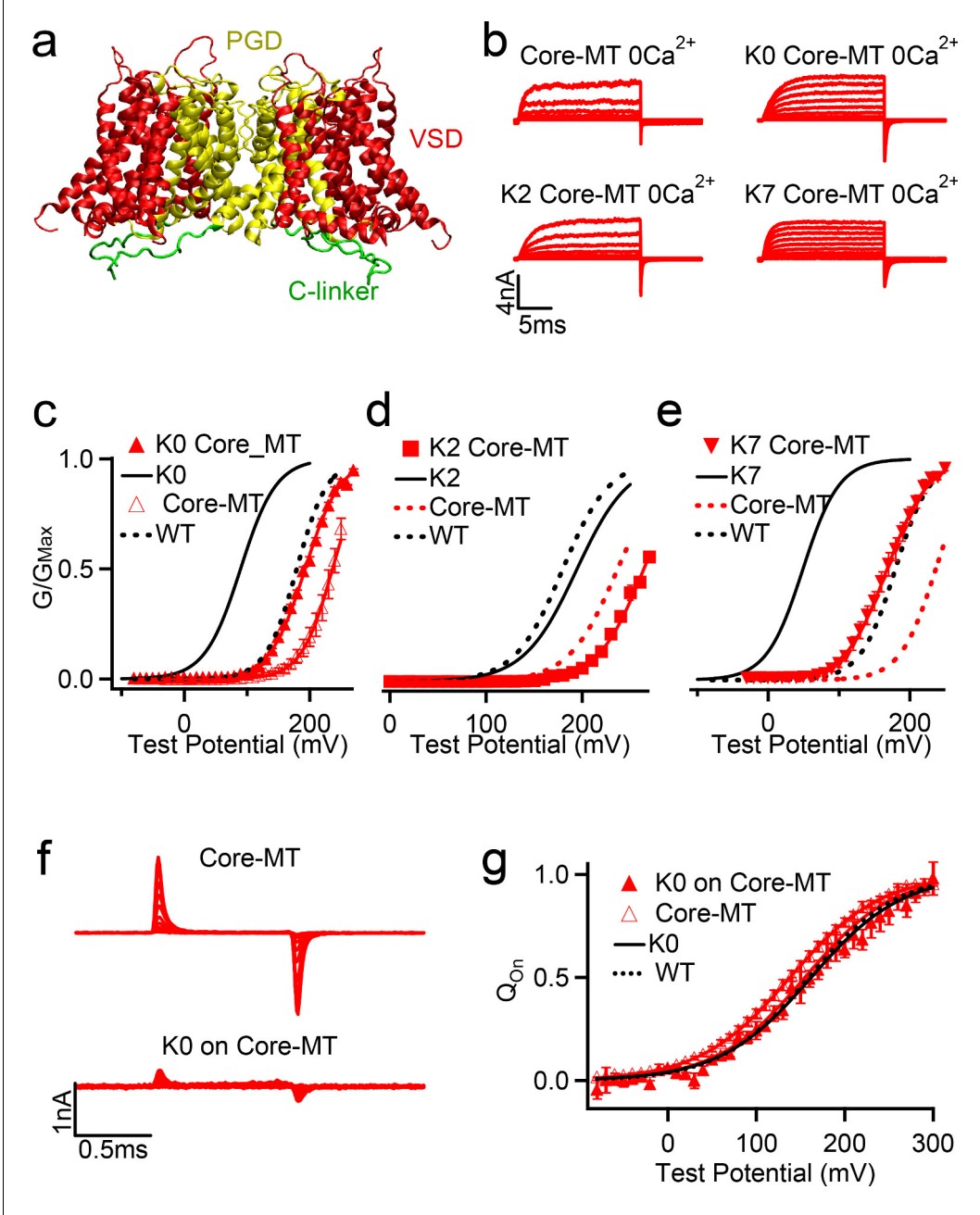

**Figure 6.** Effects of C-linker scrambling mutations of voltage activation of Core-MT BK channels. (**a**) Illustration of the structure of the Core-MT BK channel, where the whole CTD is absent. The mini-tail is omitted in the illustration for clarity. (**b**) Macroscopic currents of the WT Core-MT construct as well as the K0, K2 and K7 C-linker scrambling mutants. The currents were elicited in 0 $[Ca^{2+}]_i$ by voltage pulses from −30 to 250 mV with 20 mV increments. The voltages before and after the pulses were −50 and −80 mV, respectively. (**c-e**) G-V curves for WT Core-MT and K0, K2 and K7 mutants in 0 $[Ca^{2+}]_i$ (red traces). The G-V curves of the Full-Length channels are also shown for reference (black traces). All lines were fit to the Boltzmann relation (see Materials and methods) with $V_{0.5}$ of 235.0 ± 3.1 mV for WT Core-MT; 192.6 ± 3.8 mV for K0 Core-MT; 263.5 ± 4.0 mV for K2 Core-MT; and 167.5 ± 4.2 mV for K7 Core-MT. (**f**) Gating current traces of Core-MT and K0 in Core-MT mSlo1 channels. Voltage pulses were from −80 to 300 mV with 20 mV increments. (**g**) Normalized Q-V relation of on-gating currents. The smooth curves are fits to the Boltzmann function with a $V_{0.5}$ and slope factor of 138.0 ± 3.1 mV and 51.3 ± 2.8 mV for Core-MT. and 154.7 ± 6.8 mV and 50.1 ± 5.6 mV for K0 on Core-MT.

The online version of this article includes the following source data for figure 6:

*Figure 6 continued on next page*

*Figure 6 continued*

**Source data 1.** Data from electrophysiology experiments showing the G-V curves of the full-length and Core-MT WT, K0, K2 and K7 hSlo1 channels in 0 [Ca$^{2+}$] as depicted in *Figure 6c–e*.
**Source data 2.** Data from electrophysiology experiments showing Q-V relation of on-gating currents for full-length and Core-MT WT and K0 as depicted in *Figure 6g*.

The observation that linker sequence scrambling mutations can modulate BK activation even in the Core-MT background is remarkable, providing a direct evidence that the C-linker is more than an inert, passive covalent linker for coupling the gating ring and PGD. Instead, the linker plays a more direct and more specific role in modulating the opening of BK pore, such as through its interactions with the membrane environment. As such, the linker could be considered an integral component of the sensing apparatus.

### Removal of membrane anchoring Tyr in a C-linker BK mutant recovers WT-like gating

We note that Tyr is not the only type of residues being shuffled in the sequence scrambling (*Table 1*) and that interactions of other residues, particularly charged ones (*Tian et al., 2019*), with membrane and water could also affect the open/close equilibrium of the channel. This may explain why K1 and K2 mutant channels have different $V_{0.5}$, even though the nearest Tyr is at position 331 in both mutants (*Table 1*). To further examine if Tyr residues indeed provide the dominant contributions, we replaced Y330 with Gly in the full-length K0 mutant to completely remove the aromatic side chain. The mutant expressed robust currents (*Figure 7a*). Strikingly, K0 Y330G mutation abolished the effects of K0 on G-V relation and largely shifted the G-V back to that of the WT, with a $V_{0.5}$ of $169.8 \pm 5.0$ mV as compared to $183.4 \pm 3.2$ mV for the WT (*Figure 7b*). This, together with the correlation shown in *Figure 3*, provides a direct support that membrane anchoring effects of Tyr are mainly responsible for modulating the activation voltage in the C-linker sequence scrambling mutants.

## Discussion

We have combined atomistic simulations and experiment to determine the role of the 15-residue C-linker in the sensor-pore coupling of BK channels. As the only covalent connection between PGD and CTD of BK channels, C-linker has been widely assumed to play an important role in mediating the information transfer and domain coupling. Our analysis show that C-linker is a structured loop with extensive contacts with CTD and VSD, and remain highly stable in both metal-bound (activated) and metal-free (deactivated) states. Dynamic coupling analysis confirms that C-linker is the key pathway of the senor-pore coupling. However, the linker is not just an inert pathway. Instead, sequence scrambling of the C-linker can greatly affect the activation voltage of BK channels. Atomistic simulations revealed that the effects on the activation voltage could be mainly attributed to the nonspecific interactions between the C-linker and membrane interface, particularly membrane anchoring effects of Tyr residues. These conclusions were supported by additional experiments showing that the effects of C-linker sequence scrambling persist in the Core-MT constructs, where the gating ring is completely removed, and that replacing the key Tyr residue with Gly largely abolishes the shift in the G-V curve in a selected C-linker mutant. To the best of our knowledge, this is one of the first direct demonstrations of how nonspecific membrane interactions can modulate TM protein function.

In BK channels the cytosolic gating ring is the Ca$^{2+}$ sensor, harboring Ca$^{2+}$ binding sites (*Wei et al., 1994*; *Shi et al., 2002*; *Xia et al., 2002*; *Zhang et al., 2010*; *Hite et al., 2017*; *Tao et al., 2017*; *Tao and MacKinnon, 2019*), and important for the coupling between the voltage sensor and the gate (*Zhang et al., 2017*). The C-linker is the only covalent connection between the gating ring and the activation gate, and in this study we show that it is a major path for functional coupling between the gating ring and the pore (*Figure 2c,d*). Therefore, the C-linker is important for the coupling of both Ca$^{2+}$ and voltage sensors to the gate. We show in this study that the C-linker scrambling mutation K0 alters the voltage dependence of channel opening but not VSD activation (*Figures 1* and *6*, *Figure 2—figure supplement 1*, *Figure 2—figure supplement 2*),

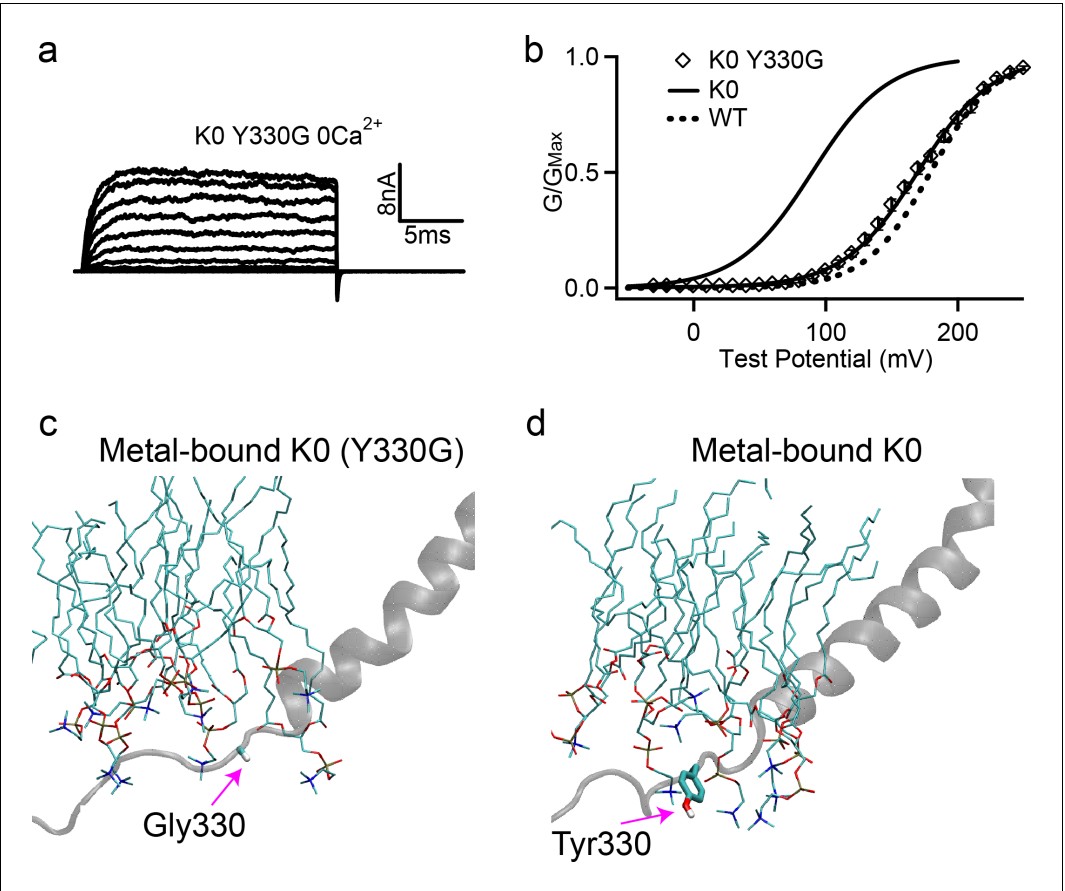

**Figure 7.** Removing Tyr sidechain in the K0 mutant full-length hSlo1 channel recovers WT-like voltage activation. (a) Macroscopic currents of K0 (Y330G) mutant channel. The currents were elicited in 0 $[Ca^{2+}]_i$ by voltage pulses from −30 to 250 mV with 20 mV increments. The voltages before and after the pulses were −50 and −80 mV, respectively. (b) The G-V curve in 0 $[Ca^{2+}]_i$, showing the recovery of activation voltage ($V_{0.5}$) of the K0 (Y330G) to the WT level. The solid line was fit to the Boltzmann relation (see Materials and methods) with 169.8 ± 5.0 mV for the K0 (Y330G). The G-V curves of the full-length WT and K0 mutant channels are also shown for reference (solid and dash lines). (c-d) Representative molecular snapshots showing the effects of Y330G mutation in K0 channels. See *Figure 4* caption for the details of molecular representations.

The online version of this article includes the following source data for figure 7:

**Source data 1.** Data from electrophysiology experiments showing G-V curves of the full-length K0 Y330G hSlo1 channels in 0 $[Ca^{2+}]_i$ as depicted in *Figure 7b*.

indicating that the mutation may alter either the activation gate or the coupling between the VSD and the pore. BK channel activation by voltage and $Ca^{2+}$ can be well described by the HA allosteric model (*Horrigan and Aldrich, 2002*; *Figure 8a*). We used this model to simulate how $V_{0.5}$ changes with either the changes of the equilibrium for the intrinsic open-closed transitions of the activation gate (L0, *Figure 8a,b*) or the changes of the allosteric coupling between the VSD and the activation gate (D, *Figure 8a,c*). The results show that either changes could approximately reproduce the $V_{0.5}$ changes caused by the C-linker scrambling mutations (*Figure 3*). These results support that the interaction of the Tyr side chain with the membrane in these C-linker scrambling mutations may affect BK channel activation by altering either the activation gate or the coupling between the VSD and the pore. However, since the C-linker is important for the coupling of the pore with both $Ca^{2+}$ and voltage sensors, we expect that the C-linker scrambling mutations would have altered both $Ca^{2+}$ and voltage-dependent activation if they were to primarily affect coupling. Since these mutations have little effect on $Ca^{2+}$ dependent activation (*Figure 3*), it seems that the Tyr-membrane interactions primarily promote gate opening. We notice that the gating ring has some influence on the functional effects of the mutations (*Figures 1* and *6*, *Table 1*), these effects may be due to the conformational

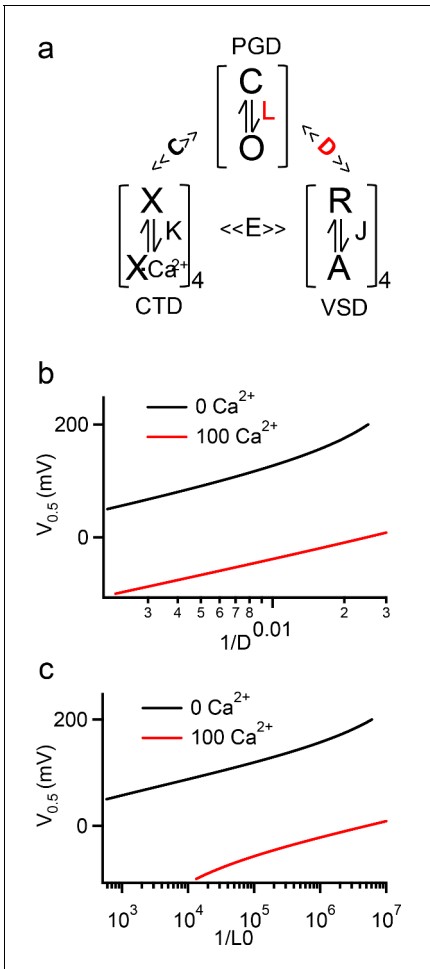

**Figure 8.** HA model simulation of $V_{0.5}$ changes with D, L0 and $[Ca^{2+}]_i$. (a) The HA model for BK channel activation, where L, J, and K are equilibrium constants to represent conformational changes in the PGD, VSD, and CTD domains respectively. D, C, and E are allosteric constants for coupling among the three domains. (b, c) Simulations of $V_{0.5}$ vs 1/D (b) and 1/L0 (c) in 0 and 100 µM $[Ca^{2+}]_i$. The HA model parameters used here are from fittings to the WT BK channels (**Zhang et al., 2014**), with $L_0 = 3.5e-07$, $Z_L = 0.18$, $Z_J = 0.59$, $V_H = 159$, $C = 10.0$, $D = 48$, $E = 4.2$ and $K_D = 18.0$. The curves are plotted with only D (b) or $L_0$ (c) varying, while all other parameters remain constant.

The online version of this article includes the following source data for figure 8:

**Source data 1.** Data for HA model simulation parameters and results as depicted in *Figure 8b,c*.

restraints imposed by the gating ring on the C-linker, but such restraints may not be sensitive to conformational changes caused by $Ca^{2+}$ binding.

Our conclusion that the C-linker is unlikely an inert component of the linker-gating ring 'passive spring' is also consistent with other structural and functional studies of BK channels. For example, the Cryo-EM structures of BK channels (**Hite et al., 2017**; **Tao et al., 2017**) reveal that the previously published poly-AAG insertion site (**Niu et al., 2004**), right after residue S337 (**Supplementary file 1**), actually locates in a short loop following the C-linker segment - $Y_{332}$GGSY$_{336}$- that forms stable contacts with RCK1 N-lobe (**Figure 1—figure supplement 2**). The inserted residues would project away from the channel (**Figure 1—figure supplement 5**), and are very unlikely to affect the effective C-linker length (or the gating ring-pore distance) as originally designed. Instead, the observed effects in BK gating $V_{0.5}$ upon insertion/deletion of C-linker residues could likely be attributed to certain nontrivial structural and/or dynamical impacts, such as weakening of the VSD/CTD interactions. Another important evidence that is inconsistent with the passive spring model comes from the study of Core-MT BK channels. Since the whole gating ring is removed, the Core-MT construct should correspond to a state where the passive spring is fully relaxed and thus $V_{0.5}$ maximizes. Yet, the $V_{0.5}$ of WT Core-MT is only ~52 mV larger than the full-length BK channel (**Table 1**). This is far below what may be expected based on poly-AAG insertion mutants, which can increase $V_{0.5}$ by ~142 mV with (AAG)$_3$ inserted (**Niu et al., 2004**) (also see **Supplementary file 1**). Interestingly, the recently published Cryo-EM structures confirms that β subunits make extensive contacts with the C-linker for influencing the channel gating and manipulating the channel function (**Tao and MacKinnon, 2019**).

TM ion channels and receptors frequently contain separate TM domains, which directly mediate function such as ion permeation, and intracellular and extracellular sensing domains, which often control the function of TM domains in response to various chemical signals (**Hille, 2001**). The general role of the covalent linkers connecting the sensing and TM domains is of great general interest (**Magleby, 2003**; **Bavro et al., 2012**). A central question is whether the linker mainly provides an inert and passive connection between sensing and TM domains or it should be considered an element of signal sensing itself. Dissecting the potential roles of covalent linkers is challenging because multiple sources of interactions and conformational transitions could impose competing strains on the linker to modulate functional regulation. It is difficult to design experiments that can unambiguously test and validate whether a particular type of strain imposed on the covalent linker could lead to predictable functional outputs. Linker sequence permutation could provide an effective strategy to suppress the potential consequence of specific (but unknown) linker interactions, allowing one to test the functional role of a single type of strain

imposed on the linker (such as membrane anchoring). Our findings show that non-specific interactions of the C-linker can regulate BK voltage gating. Therefore, covalent linkers of membrane proteins could serve as sensors of signals that perturb their interaction with the environment, which in turn can modulate the functional center in the TM domain, may it be the gate of ion channels or intramolecular signaling pathway in receptors.

# Materials and methods

## Key resources table

| Reagent type (species) or resource | Designation | Source or reference | Identifiers | Additional information |
|---|---|---|---|---|
| Gene *Mus musculus* | mslo1 | | GenBank GI: 47143 | |
| Gene *Mus musculus* | Core-MT | Dr. Lawrence Salkoff | | PMID:24067659 |
| Biological sample (*Xenopus laevis*) | oocyte | *Xenopus laevis* | | *Xenopus laevis* purchased from Nasco, Fort Atkinson, WI |
| Commercial assay or kit | mMESSAGE T7 Transcription Kit | Thermo Fisher | AM1344 | |
| Software, algorithm | Igor Pro 4.0 | WaveMetrics | https://www.wavemetrics.com/products/igorpro | |
| Sequence-based reagent | For site-directed mutagenesis | This paper | PCR primers | PCR primers seq for mutations made in this study (each mutation utilized two primers: b and c).<br>K0<br>b: gCtctGCTgTActtGgaCCCc TtgccgTaGCGGTTTCCTATTAACTC<br>c: cCaagTAcAGCagaGcTgtc tccAagggGCACATTGTAGTCTGTG<br>K1<br>b: cTtGtAcgaCCCGccaTaCTT gttGccgatCcTTAACTCTATGATTTCAG<br>c: ggCGGGtcgTaCaAgAGCGCtGT ccGcaagAGcggGCACATTGTAGTCTG<br>K2<br>b: gagTAGctaCCgCCcTtgTagTT cttgcgTCCTATTAACTCTATGATTTC<br>c: gGGcGGtagCTActcCGC cagggtctcAgGAAAGCACATTGTAGTC<br>K4<br>b: ctTAgcGGAGtaCgaGccTccg TaGtttcTTCCTATTAACTCTATGATTT<br>c: gCtcGtaCTCCgcTAagaaGGTTAG gaGcAaAggGCACATTGTAGTCTGT<br>K7<br>b: CTAacGGcGCtgtaGctTccCcc GtaGTTTCCTATTAACTCTATG<br>c: gCtacaGCgCCgtTAGgaaGag TaagGGAAGAAAGCACATTGTAG<br>K0 on Core-MT<br>b: gCtctGCTgTActtGgaCCCcT tgccgTaGCGGTTTCCTATTAACTC<br>c: cCaagTAcAGCagaGcTgtctccA agggtGGAGTCAAGGAATCATTA<br>K7 on Core-MT<br>b: gaaGagTaagGGAAGAAAGGGAGTCAAG<br>c: GAAGAAAGGGAGTCAAGGAATCAT<br>K2 on Core-MT<br>b: CCTTGACTCCtTTTCcTgagaccctgG<br>c: gGAAAaGGAGTCAAGGAATCATTATG<br>K0 Y330G<br>b: GCCGccGCGGTTTCCTATTAACTC<br>c: GAAACCGCggCGGCAAGGGGTCCAAG |

## Homology modeling and atomistic simulations

As described previously (*Jia et al., 2018*), homology models of the WT metal-bound and metal-free hSol1 channels were built using Modeller v9.14 (*Sali and Blundell, 1993*) based on sequence alignment of *Tao et al., 2017*. Sequence alignment shows 55.95% identity for the full-length channels. The sequence identities in PGD is higher at 61.96%. The high level of sequence identity suggests that the homology models are likely reliable. This has been confirmed by direct comparison with the recently published Cryo-EM structures of hSlo1 (*Tao and MacKinnon, 2019*; *Figure 1—figure supplement 3*). The backbone RMSD between the model and new structure is only 2.18 Å at the whole channel level and as low as 0.87 Å in the PGD. Structures for C-linker scrambling mutants were build based on the WT hSlo1 models using CHARMM (*Brooks et al., 2009*).

Using CHARMM-GUI server (*Lee et al., 2016*), the homology modeled hSlo1 structures were inserted in POPC lipid bilayers followed by solvation using the TIP3P water model (*Jorgensen et al., 1983*). 450 mM KCl was then added, same as used in Cryo-EM structure determination (*Hite et al., 2017*; *Tao et al., 2017*). $K^+$ ions were added to each binding site of the selectivity filter (S1-S4) without any intervening water. Each system was first energy minimized, followed by multiple cycles of equilibration dynamics with gradually decreasing harmonic restrains on positions of selected protein/lipid heavy atoms. To ensure that the size of the simulation box become stable, in the last equilibration step, only protein heavy atoms were harmonically restrained while the system equilibrated under NPT (constant particle number, pressure and temperature) condition. The final simulation box size is ~18×18 x 15.4 $nm^3$ with ~476,000 atoms, containing ~97,000 water molecules and ~800 lipid molecules. The Charmm36m all-atom force field (*Huang et al., 2017*) was used for all systems. The production simulations were performed using CUDA-enabled Amber14 (*Case et al., 2014*). The MD time step was set at two fs. Particle Mesh Ewald (PME) algorithm (*Darden et al., 1993*) with a cutoff at 12 Å was used to describe the electrostatic interactions. Van der Waals interactions were cutoff at 12 Å with a smooth switching function starting from 10 Å. Lengths of all covalent bonds involving hydrogen atoms were fixed using SHAKE algorithm (*Ryckaert et al., 1977*). The system temperature was maintained at 298 K by Langevin dynamics with a friction coefficient of 1 $ps^{-1}$, and the pressure was maintained semi-isotropically at 1 bar at both x and y (membrane lateral) directions using the Monte-Carlo barostat method (*Chow and Ferguson, 1995*; *Åqvist et al., 2004*). Three independent 800-ns NPT production simulations were performed for each construct (WT, K0, K2 and K7) in both metal-free and bound states, with an aggregated simulation time of 19.2 μs. Snapshots were saved every 50 ps for post analysis.

## Structural and dynamic analysis

All analyses were performed using a combination of in-house scripts, MDAnalysis (*Michaud-Agrawal et al., 2011*) and Gromacs2016 (*Hess et al., 2008*; *Abraham et al., 2015*) software. Only snapshots from the last 150 ns of all production MD trajectories were used for the calculation of Tyr-membrane interactions (SASA of burial, hydrogen bonding, π-cation and carbon-carbon contacts) as well as RMSF and the C-linker contact map (SI). A (hydrophobic) carbon-carbon contact was considered formed if the distance is no greater than 4.5 Å. The π-cation interaction was identified when the distance between the center of mass of the Tyr aromatic ring and Nitrogen atom of POPC choline group is no greater than 5.0 Å. We note that nonpolarizable fore fields such as Charmm36m used in this work have been shown to be capable of accurately describing the geometries of π-cation interactions, even though the energetics could be improved using modified Lennard-Jones potentials (*Khan et al., 2019*). Similarly, the cutoff was set at 5.0 Å for calculation of the C-linker contact map (SI) while only the heavy atoms of each residue were considered. Hydrogen bonds were analyzed using the MDAnalysis 'HyrogenBondAnalysis' class with default criteria. The number of pore water molecules was calculated using the same criteria as described previously (*Jia et al., 2018*). Dynamic network analysis was performed using the *Networkview* (*Eargle and Luthey-Schulten, 2012*) plugin of VMD (*Humphrey et al., 1996*). For this, snapshots were extract every one ns from the 800 ns molecular dynamic trajectories. To build the network, each amino acid was represented as a single node at their Cα position and a contact (edge) was defined between two nodes if the minimal heavy-atom distance between them was within a cutoff distance (4.5 Å) during at least 75% of the trajectory. The resulting contact matrix were then weighted based on the correlation coefficients of dynamic fluctuation ($\underline{C}_{ij}$), calculated using the Carma software (*Glykos, 2006*), as $w_{ij}$ = -log

($|C_{ij}|$), where $C_{ij} = <\Delta r_i(t).\Delta r_j(t)> / (<\Delta r_i(t)^2> <\Delta r_j(t)^2>)^{1/2}$ and $\Delta r_i(t)=r_i(t) - < r_i(t)>$, $r_i(t)$ is the position of the atom corresponding of the $i^{th}$ node and $<>$denotes ensemble average (over the MD trajectory). The path length between the desired nodes were then calculated as the sum of the edge weights. The shortest (optimal) path, calculated using Floyd-Warshall algorithm (*Floyd, 1962*), is believed to represent the dominant mode of communication. Slightly longer (suboptimal) paths were also calculated. VMD was used for preparing all molecular illustrations.

## Mutations and expression

Mutations in all experiments were made by using overlap-extension PCR (polymerase chain reaction) with Pfu polymerase (Stratagene) from the mbr5 splice variant of *mslo1* (Uniprot ID: Q08460) (*Butler et al., 1993*). And then all PCR-amplified regions were verified by sequencing (*Shi et al., 2002*). RNA was then transcribed in vitro from linearized DNA with T3 polymerase (Ambion, Austin, TX) and an amount of 0.05–50 ng/oocyte RNA was injected into oocytes (stage IV-V) from female *Xenopus laevis*. After injection, these oocytes were incubated at 18°C for 2–7 days.

## Electrophysiology

We used inside-out patches to record Ionic currents with an Axopatch 200-B patch-clamp amplifier (Molecular Devices, Sunnyvale, CA) and ITC-18 interface and Pulse acquisition software (HEKA Elektronik GmbH, Holliston, MA). Borosilicate pipettes those were used for inside-out patches had 0.5–1.5 MΩ resistance and then formed patches from oocyte membrane. The current signals were recorded at 50 KHz sample rate (20-μs intervals) with low-pass-filtered at 10 KHz. In order to remove capacitive transients and leak currents, we applied a P/4 protocol with a holding potential of –120 mV. Our solutions used in recording ionic currents were listed below. 1) Pipette solution (in mM): 140 potassium methanesulphonic acid, 20 HEPES, 2 KCl, 2 MgCl$_2$, pH 7.2. 2) The nominal 0 μM [Ca$^{2+}$]$_i$ solution, which contained about 0.5 nM free [Ca$^{2+}$] $_i$ (in mM): 140 potassium methanesulphonic acid, 20 HEPES, 2 KCl, 5 EGTA, and 22 mg/L (+)−18-crown-6-tetracarboxylic acid (18C6TA), pH 7.2. 3) Basal bath (intracellular) solution (in mM): 140 potassium methanesulphonic acid, 20 HEPES, 2 KCl, 1 EGTA, and 22 mg/L 18C6TA, pH 7.2. Then we added CaCl$_2$ into basal solution to obtain the desired free [Ca$^{2+}$]$_i$, which was determined by a Ca$^{2+}$-sensitive electrode (Thermo Electron, Beverly, MA). Gating currents were recorded from inside-out patches at 200 kHz sample rate and 20 kHz low-pass filtration with leak subtraction using a -P/4 protocol. The pipette solution contained (in mM): 127 tetraethylammonium (TEA) hydroxide, 125 methanesulfonic acid, 2 HCl, 2 MgCl$_2$, 20 HEPES, pH 7.2, and the Internal solution contained (in mM): 141 N-methyl-D-glucamine (NMDG), 135 methanesulfonic acid, 6 HCl, 20 HEPES, 5 EGTA, pH 7.2. All chemicals were from Sigma-Aldrich unless otherwise noted, and all the experiments were done at room temperature (22–24°C).

## Electrophysiology data analysis

Relative conductance (G) was obtained by measuring macroscopic tail current at –80 mV or −120 mV. The conductance-voltage (G-V) relationships was plotted to fit with the Boltzmann function:

$$G/G_{Max} = 1/(1 + exp(-ze_o(V - V_{0.5})/kT)) = 1/(1 + exp((V_{0.5} - V)/b)) \tag{1}$$

Where $G/G_{Max}$ means the ratio of conductance to maximal conductance, $z$ means the number of equivalent charges, $e_o$ means the elementary charge, $V$ means membrane potential, V$_{0.5}$ means the voltage where $G/G_{Max}$ reaches 0.5, $k$ means Boltzmann's constant, $T$ means absolute temperature, and $b$ means the slope factor with units of mV. Each G-V relationship was the average of 3–15 patches and error bars in the figures is standard error of means (SEM.).

## Model simulation of electrophysiology data

HA model (*Horrigan and Aldrich, 2002*) was used to simulate the relationship between V$_{0.5}$ and D or L$_0$ in *Figure 8*.

$$Po = L(1 + KC + JD + JKCDE)^4/(L(1 + KC + JD + JKCDE)^4 + (1 + K + J + JKE)^4) \tag{2}$$

V$_{0.5}$ is the voltage for Po = 0.5, and the *Equation (2)* changes to:

$$L(1 + KC + JD + JKCDE)^4 = (1 + K + J + JKE)^4 \tag{3}$$

where

$$L(V_{0.5}) = L_0 exp(-Z_L V_{0.5}/KT) \tag{4}$$

$$J(V_{0.5}) = J_0 exp(-Z_J V_{0.5}/KT) = exp((V_{0.5} - V_H)Z_J V/KT) \tag{5}$$

$$K = [Ca^{2+}]_i/K_D \tag{6}$$

The parameters for the HA model were obtained previously by fitting the model to WT BK data (*Zhang et al., 2014*), where $L_0$ = 3.5e-07, $Z_L$ = 0.18, $Z_J$ = 0.59, $V_H$ = 159, C = 10.0, D = 48, E = 4.2 and $K_D$ = 18.0. To simulate $V_{0.5}$ vs 1/D or $V_{0.5}$ vs $1/L_0$, only D or $L_0$ varied while all other parameters remained constant.

# Acknowledgements

We thank Rohit Pappu for the original design of C-linker mutants. This work was supported by National Institutes of Health Grants R01 HL142301 and GM114300 (to Chen). We also would like to thank Roderick MacKinnon and Xiao Tao for sharing the hSlo1 Cryo-EM structures prior to their release in PDB. Computing for this project was performed on the Pikes cluster housed in the Massachusetts Green High-Performance Computing Center (MGHPCC).

# Additional information

### Funding

| Funder | Grant reference number | Author |
| --- | --- | --- |
| National Institute of General Medical Sciences | GM114300 | Jianhan Chen |
| National Heart, Lung, and Blood Institute | HL142301 | Jianhan Chen |

The funders had no role in study design, data collection and interpretation, or the decision to submit the work for publication.

### Author contributions

Mahdieh Yazdani, Guohui Zhang, Data curation, Formal analysis, Visualization, Writing - original draft; Zhiguang Jia, Data curation, Formal analysis, Visualization; Jingyi Shi, Resources; Jianmin Cui, Conceptualization, Funding acquisition, Methodology, Project administration, Writing - review and editing; Jianhan Chen, Conceptualization, Formal analysis, Funding acquisition, Methodology, Writing - original draft, Project administration, Writing - review and editing

### Author ORCIDs

Mahdieh Yazdani (iD) https://orcid.org/0000-0003-1300-4599
Guohui Zhang (iD) https://orcid.org/0000-0001-6905-0637
Jianmin Cui (iD) https://orcid.org/0000-0002-9198-6332
Jianhan Chen (iD) https://orcid.org/0000-0002-5281-1150

### Decision letter and Author response

Decision letter https://doi.org/10.7554/eLife.55571.sa1
Author response https://doi.org/10.7554/eLife.55571.sa2

## Additional files

### Supplementary files

• Supplementary file 1. Effects of C-linker insertion/deletion mutations on the voltage required for half channel activation ($V_{0.5}$) of hSlo1 in absence of $Ca^{2+}$ and $Mg^{2+}$ (data extracted from: *Niu et al., 2004*). The position of insertion (S337) is colored red.

• Transparent reporting form

### Data availability

All data generated or analysed during this study are included in the manuscript and supporting files.

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
