## [Decision Letter]

**Acceptance summary:**

Gating, the closed to open state transition in high conductance potassium (BK) ion channels is modulated by complex interactions with a specialized domain that acts as a calcium sensor. In this manuscript Yazdani et al. present a set of experiments and simulation results which demonstrate that the BK, high-conductance channel, relies on the physical characteristics of a linker region joining the calcium-sensing domains to the pore domain to allosterically transmit energy for gating. The authors have used a nice combination of theory and experiments to provide solid evidence for this interaction. these results represent an advance in our understanding of these important proteins. After review, the authors have nicely revised their manuscript.

**Decision letter after peer review:**

Thank you for sending your article entitled "Nonspecific membrane interactions can modulate BK channel activation" for peer review at *eLife*. Your article has been evaluated by three peer reviewers, one of whom is a member of our Board of Reviewing Editors, and the evaluation has been overseen by Richard Aldrich as the Senior Editor.

Apart from the individual reviews appended below, several concerns were raised by reviewers during online consultation. Mainly, the paper needs to be completely rewritten to better integrate simulations with experiments. Also, the reviewers think that several new experiments are needed to substantiate the conclusion, especially those pertaining the experimental part of the manuscript. Given the current world crisis due to COVID-19 and the closing of several labs, we want to gage the possibility that you can perform the experiments in reasonable time.

Reviewer #1:

The manuscript by X presents both experimental and computational (molecular dynamics) data pertaining the role of the C-linker, in BK channel activation. This region connecting the transmembrane domains and the intracellular ligand-binding domains has been thought of as a passive spring transmitting conformational changes from the RCK domains to the pore region and perhaps the voltage sensor. In this work, the authors show that tyrosines in the C-linker form non-specific interactions with inner membrane lipids and these interactions are capable of regulating the range of voltage over which channel activation occurs. This is an interesting result, and interactions of aromatic residues with the membrane may be found to be a general principle in other channels. My comments are the following:

– The use of only the V1/2 value as a reporter of the open-close equilibrium is very limited. The mutations in the c-linker could affect the allosteric interactions between the pore, VSD and RCK domains in ways that are not predictable. The authors should provide other experimental measurements. Since the manuscript wants to study the effect of the composition of the c-linker on coupling of ligand to channel opening, why are all the experiments carried out in the absence of Ca^2+^? It seems that all the conclusions regarding coupling are being derived from the MD simulations, but no effort is made to actually dissect the effect of linker composition on coupling. The authors should show data on the dependence of v1/2 on calcium concentration of their constructs.

– The manuscript has several grammatical errors and typos that need to be corrected. For example, “bended”.

– Describing the C-linker as a passive spring is an over simplification, the amino acid composition , even if not forming specific interactions with other domains can affect the mechanical properties of the linker in a sequence-specific manner. The authors should be careful in stating this.

– In general, calling the interaction of a tyrosine residue with the membrane non-specific seems non satisfactory. The interaction is specific to the tyr side chain and is mediated by specific physical-chemistry.

– When the authors refer to Figure 5 in the subsection “Tyr membrane anchoring affects BK channel activation similarly without the gating ring”, it should actually be Figure 6.

– Figure 2 legend. The authors refer to the small values of RMSF as indicating limited flexibility of the C-linker. However, RMSF is not a measure of flexibility in a mechanical sense.

– Figure 2 legend. There are no panels E and F in the figure.

– Subsection “C-linker is a structured loop with limited flexibility and a key dynamic pathway of BK gating ring-pore coupling”, solubility of C-linker. How do you define solubility in this context, since the linker is attached to the protein? Perhaps using solvation of the linker is more precise.

– "These structures reveal that the C-linker is a structured loop with essentially identical conformations in both metal-free (closed) and metal-bound (open) states (Figure 1B)." Please show a more detailed figure of the linker in both states, since the figure as is does not bear out this information.

– Figure 2—figure supplement 6, what is the correlation coefficient for? It seems that you cannot compare the simulation shown in Figure 2—figure supplement 6 with the Niu et al. paper. Also, if you remove K0 from the graph, it seems you get a good correlation linear correlation as expected for a spring.

Reviewer #2:

Through molecular dynamics simulations, Yazdani et al. suggest that the C-linker segment connecting S6 and the gating ring domain is a relatively stable loop. The simulation results are accompanied by electrophysiological measurements from WT and select C-linker mutants.

The authors suggest that if the first (most N-terminal) Tyr residue in the C-linker segment is moved towards S6 (or upstream), the Tyr residue can interact with membrane lipids (I think this is what the authors call "nonspecific membrane interactions"). Such interactions are mostly absent or rare in wild-type channels presumably because the Tyr residue is too far away.

The simulation results seem straight forward and solid with three independent systems constructed (but see my comments elsewhere). The molecular dynamics is a useful tool but the method is imperfect. (Like other simulation results,) some concerns do exist – but I think that readers can decide on their own.

In an ideal study, there should be a high level of integration between the simulation and experimental results. I cannot say that this study in the present form achieves such a level. The study shows that what a single Tyr residue could achieve when selectively or strategically positioned, and, perhaps indirectly, tells us about how WT channels work.

Probably two of my biggest concerns are the terms "inert" and "nonspecific" as used in this manuscript as described elsewhere.

Reviewer #3:

The C-linker physically couples the cytosolic and transmembrane domains in the large-conductance potassium (BK) channel yet its mechanistic role is not fully understood. In the present work, Cui, Cheng and co-workers aim at characterizing the C-linker's role in BK activation by investigating a series of linker sequences. The main conclusion is that the C-linker serves as a sensor of environmental signals that modulates BK channels by means of membrane-mediated nonspecific interactions. I believe the manuscript is of relevance and I would like to see it published provided that revisions are addressed by the authors.

Because the C-linker is a structured loop with an equilibrium end-to-end distance of ~ 31 Å and with extensive amino-acid interactions with the cytosolic and voltage-sensor domains, the intuitive hypothesis raised by the authors is that the C-linker may impact BK activation as being part of the gating apparatus of the channel. While the modulatory nature of the C-linker was experimentally shown to be true in terms of activation-voltage shifts, no explicit rationale was proposed by the authors to account for that fact, apart from the membrane-mediated nonspecific interactions – i.e., nonspecific interactions of the linker with membrane alone could directly modulate BK activation. Although the conclusion is sound and more importantly supported by the data, one may not avoid wondering about an alternative or complementary mechanism not yet explored or discussed by the authors. It is well known that the lipid-phosphoryl moiety helps stabilizing the down state of the voltage sensor via salt-bridge interactions with the positively gating charges of the domain (see for instance, Treptow and colleagues PNAS 2011, 2012 and 2014). Accordingly, the linker constructs might affect that specific lipid-protein interaction given their "decoration" with one or more basic amino acids in addition to the membrane-anchoring tyrosine – as shown in Figure 4 and Figure 2—figure supplement 1, the linker appears to interact with the sensor domain, in all of its forms.

All said, I would then recommend the authors to analyze carefully if C-linker charges are "competing" for phospholipids sites and destabilizing salt-bridge interactions with the voltage sensor. Still by means of membrane-anchoring effects, it would not be very surprising if C-linker modulates the activation voltage of the channel by interfering with the voltage sensor.

Other points:

It was stated in Figure 1—figure supplement 4 that "RMSF and RMSD were obtained from the 150ns MD simulations with time step 0.05ns" – while understanding standard calculations in the context of MD simulations it is still not clear for me the purpose of that time step in the computation of the RMSF or RMSD adopted by the authors.

What is the purpose of the dynamical-coupling analysis shown in Figure 2 – what is the hypothesis behind that study? Is there any control or reference benchmark for the analysis – for instance, how "strongly" does the C-linker couple S6/RCK1 versus VSD/RCK1 or any other region?

The residue contact map shown in Figure 1—figure supplement 4 is very busy and does not clearly illustrate neither the orientation of the conserved residues (Y332, Y336, R329, K330, K331, R342 and K343) of the C-linker nor its packing with the VSD/RCK1 domains.

An equilibrium end-to-end distance of ~ 31 Å appears to be the necessary structural condition for a functional C-linker that couples cytosolic and transmembrane domains of the channel. Indeed, as shown in Figure 2—figure supplement 3, that structural condition appears to be shared by the wild-type and the alternative C-linker constructs (K0, K2 and K7) despite their distinct impact on channel's gating and activation voltages. If that understanding is correct then the lack of correlation between end-to-end distances and voltage-shifts induced by mutations in Figure 2—figure supplement 6 is expected and it cannot be used as an argument against the passive spring model.

Notation of figures are degenerate – for instance, there are two Supplementary Figure 3s in the text.

---

## [Author Response]

Reviewer #1:My comments are the following:– The use of only the V1/2 value as a reporter of the open-close equilibrium is very limited. The mutations in the c-linker could affect the allosteric interactions between the pore, VSD and RCK domains in ways that are not predictable. The authors should provide other experimental measurements. Since the manuscript wants to study the effect of the composition of the c-linker on coupling of ligand to channel opening, why are all the experiments carried out in the absence of Ca^2+^? It seems that all the conclusions regarding coupling are being derived from the MD simulations, but no effort is made to actually dissect the effect of linker composition on coupling. The authors should show data on the dependence of v1/2 on calcium concentration of their constructs.

We would like to thank the reviewer for raising these important questions. We have now included two lines of experimental data and some model simulations in the revised manuscript to further support that the C-linker mutations studied in this work seem not to alter voltage sensor activation but to perturb the pore opening. These new data include gating current measurements (Figures 1 and 6), G-V curve shifts by the mutations in 100 µM [Ca] (Figure 3), and simulations of the V_0.5_ changes using the HA model (Figure 8). Gating current measurements suggest that the C-linker mutations may not alter VSD activation, suggesting that these mutations may alter pore opening or the VSD-pore coupling. The G-V shifts in 0 and 100 µM [Ca[ suggest that the C-linker mutations do not alter Ca dependent activation. Since the C-linker mutations does not seem to alter the coupling between Ca binding and pore opening, we propose that these mutations may alter pore opening.

– The manuscript has several grammatical errors and typos that need to be corrected. For example, “bended”.

Thanks. The word “bended” is replaced with “bending” or “bent” when appropriate.

– Describing the C-linker as a passive spring is an over simplification, the amino acid composition , even if not forming specific interactions with other domains can affect the mechanical properties of the linker in a sequence-specific manner. The authors should be careful in stating this.

We agree with the notion that the linker sequence can affect mechanical properties without forming specific interactions with other domains. We actually examined this possibility, by simulating the free linker peptides and analyze their inherent conformational preferences. Some of the key results are summarized and discussed in the manuscript (Figure 2—figure supplement 6; subsection “C-linker scrambling mutations minimally perturb channel structure, dynamics and sensorpore coupling”, second paragraph). We conclude that the inherent mechanical properties of the linker cannot adequately explain our experimental data. Along this line, we agree that the term “passive spring” is an oversimplification. However, we retain this wording in the revised manuscript as this was the original term used to describe the previous model (Niu, Qian et al. 2004).

– In general, calling the interaction of a tyrosine residue with the membrane non-specific seems non satisfactory. The interaction is specific to the tyr side chain and is mediated by specific physical-chemistry.

We agree with that there is a specific physical nature of the interaction between Trysine sidechain and membrane/water environment. However, it seems to be true that for any given type of interactions there is a specific physical nature. By “nonspecific”, we are contrasting the nature of the C-linker/membrane interactions studied in this work with two kinds of “specific” interactions that are commonly invoked in molecular biophysics, 1) binding of one or more lipid molecules to a well-defined pocket on the channel protein in a specific conformation, or 2) specific protein interactions between two defined residues (e.g., specific contacts between the C-linker residues and other specific residues of the rest of the channel in VSD or RCK domains). The tyrosine/membrane interface interactions here are dynamic and do not involve one or a few defined configurations or any specific lipid molecules. Along this line, we respectfully argue that the wording “nonspecific” is appropriate. We have added this clarification to the revised manuscript (Introduction, last paragraph). However, to avoid any ambiguity that might arise before the “nonspecific” is clarified, we change the title to “Aromatic interactions to Membrane Modulate BK Channel Activation”

– When the authors refer to Figure 5 in the subsection “Tyr membrane anchoring affects BK channel activation similarly without the gating ring”, it should actually be Figure 6.

Thank you. This typo is corrected.

– Figure 2 legend. The authors refer to the small values of RMSF as indicating limited flexibility of the C-linker. However, RMSF is not a measure of flexibility in a mechanical sense.

Thank you. The typo is corrected

– Figure 2 legend. There are no panels E and F in the figure.– Subsection “C-linker is a structured loop with limited flexibility and a key dynamic pathway of BK gating ring-pore coupling”, solubility of C-linker. How do you define solubility in this context, since the linker is attached to the protein? Perhaps using solvation of the linker is more precise.

Good point. The word “solubility” was changed to “solvation”

– "These structures reveal that the C-linker is a structured loop with essentially identical conformations in both metal-free (closed) and metal-bound (open) states (Figure 1B)." Please show a more detailed figure of the linker in both states, since the figure as is does not bear out this information.

We note that Figure 1B does show the overlapping of the C-linker in the open and closed state. Figure 2—figure supplement 3C further compare the C-linker conformations in both states for selected C-linker mutants. More quantitative analysis (RMSF, RMSD, C-linker length) further mapped out the detailed conformational properties of the C-linker.

– Figure 2—figure supplement 6, what is the correlation coefficient for? It seems that you cannot compare the simulation shown in Figure 2—figure supplement 6 with the Niu et al. paper. Also, if you remove K0 from the graph, it seems you get a good correlation linear correlation as expected for a spring.

Figure 2—figure supplement 6. The correlation coefficient shows the *poor* correlation between the length of the *free* C-linker scrambled mutants and that predicted from the “passive spring” model (red line). The correlation coefficient remains very small (R^2^ <<0.1) even if K0 is deleted.

Reviewer #2:[…] The simulation results seem straight forward and solid with three independent systems constructed (but see my comments elsewhere). The molecular dynamics is a useful tool but the method is imperfect. (Like other simulation results,) some concerns do exist – but I think that readers can decide on their own.In an ideal study, there should be a high level of integration between the simulation and experimental results. I cannot say that this study in the present form achieves such a level. The study shows that what a single Tyr residue could achieve when selectively or strategically positioned, and, perhaps indirectly, tells us about how WT channels work.

We appreciate the important issue raised by the reviewer. We include more experimental data in the revised manuscript (**Figures 1, 3, 6**, see responses to reviewer #1, comment 1) and simulations of the changes in BK channel gating using the HA model (**Figure 8**). These new data provide a more complete account of the changes in BK channel gating mechanism as the result of the Tyr interactions with the membrane and a better integration between the simulation and experimental results.

Probably two of my biggest concerns are the terms "inert" and "nonspecific" as used in this manuscript as described elsewhere.

Please see below regarding the terms “inert” and “nonspecific”.

Reviewer #3:Because the C-linker is a structured loop with an equilibrium end-to-end distance of ~ 31 Å and with extensive amino-acid interactions with the cytosolic and voltage-sensor domains, the intuitive hypothesis raised by the authors is that the C-linker may impact BK activation as being part of the gating apparatus of the channel. […] Accordingly, the linker constructs might affect that specific lipid-protein interaction given their "decoration" with one or more basic amino acids in addition to the membrane-anchoring tyrosine – as shown in Figure 4 and Figure 2—figure supplement 1, the linker appears to interact with the sensor domain, in all of its forms.All said, I would then recommend the authors to analyze carefully if C-linker charges are "competing" for phospholipids sites and destabilizing salt-bridge interactions with the voltage sensor. Still by means of membrane-anchoring effects, it would not be very surprising if C-linker modulates the activation voltage of the channel by interfering with the voltage sensor.

We thank the reviewer for raising an interesting possibility. We further analyzed our simulation results and carefully checked the contacts of the C-linker residues with that of gating charges in the VSD domain. Our contact map analysis (Figure 2—figure supplement 1) shows that the gating charges of the VSD domain (R207, R210 and R213) do not have any contact with the C-linker residues for any construct (WT and mutants) in either metal-bound or free state. A discussion has been added to the revised manuscript (subsection “C-linker scrambling mutations minimally perturb channel structure, dynamics and sensorpore coupling”, last paragraph).

Importantly, as detailed in our response to reviewer 1, major question 1, we have now included additional experimental data to validate that the linker mutants do not affect the VSD movement. Combining atomistic simulations and experimental results, we further established that the shift in the pore open-close equilibrium is mainly due to the membrane anchoring effects of tyrosine residues.

Other points:It was stated in Figure 1—figure supplement 4 that "RMSF and RMSD were obtained from the 150ns MD simulations with time step 0.05ns" – while understanding standard calculations in the context of MD simulations it is still not clear for me the purpose of that time step in the computation of the RMSF or RMSD adopted by the authors.

The sentence has been revised to “RMSF and RMSD were obtained from the 150ns MD simulations with snapshots taken every 0.05 ns” for clarity.”

What is the purpose of the dynamical-coupling analysis shown in Figure 2 – what is the hypothesis behind that study? Is there any control or reference benchmark for the analysis – for instance, how "strongly" does the C-linker couple S6/RCK1 versus VSD/RCK1 or any other region?

We used dynamic network analysis to search for potential allosteric pathway between the CTD and PGD domains and to test if the C-linker plays a role in coupling of the two domains. The method first constructs a network with edge lengths given by negative logarithm of the covariance of dynamic fluctuation derived from MD trajectories, w_ij_ = -log (|C_ij_|). The “strongest” pathway of dynamic coupling between two remote locations is then given as the shortest path of the network (see Materials and methods). The analysis is generally not applicable to compare or evaluate the relative strengths of coupling between different domains.

The residue contact map shown in Figure 1—figure supplement 4 is very busy and does not clearly illustrate neither the orientation of the conserved residues (Y332, Y336, R329, K330, K331, R342 and K343) of the C-linker nor its packing with the VSD/RCK1 domains.

Indeed, the plot shown in Figure 1—figure supplement 4B is meant to provide a 2D illustration of how the protein coordinate a ligand (C-linker here). It does not show the full 3D placement or orientation of various residues to avoid overlaps, but is generally considered a good compromise in order to show all contacts with some spatial information.

An equilibrium end-to-end distance of ~ 31 Å appears to be the necessary structural condition for a functional C-linker that couples cytosolic and transmembrane domains of the channel. Indeed, as shown in Figure 2—figure supplement 3, that structural condition appears to be shared by the wild-type and the alternative C-linker constructs (K0, K2 and K7) despite their distinct impact on channel's gating and activation voltages. If that understanding is correct then the lack of correlation between end-to-end distances and voltage-shifts induced by mutations in Figure 2—figure supplement 6 is expected and it cannot be used as an argument against the passive spring model.

As explained in response to reviewer 2 above, the studies of the inherent conformational properties of isolated linker peptides were to evaluate if they could be used to explain the observed V_0.5_ shifts. The manuscript has been revised to add a discussion similar to above to help clarify the purpose of this analysis (see subsection “C-linker scrambling mutations minimally perturb channel structure, dynamics and sensorpore coupling”, second paragraph).

Notation of figures are degenerate – for instance, there are two Supplementary Figure 3s in the text.

Thank you for pointing this out. This typo has been fixed in both SI and the main text.